# Experimental Investigation of Air Quality in a Subway Station with Fully Enclosed Platform Screen Doors

**DOI:** 10.3390/ijerph17145213

**Published:** 2020-07-19

**Authors:** Liping Pang, Chenyuan Yang, Xiaodong Cao, Qing Tian, Bo Li

**Affiliations:** 1School of Aeronautic Science and Engineering, Beihang University, Beijing 100191, China; pangliping@buaa.edu.cn (L.P.); m15235135340@163.com (C.Y.); 2School of Aero-Engine, Shenyang Aerospace University, Shenyang 110136, China; 3School of Information Science and Technology, North China University of Technology, Beijing 100144, China; tianqing@ncut.edu.cn (Q.T.); 18811756298@163.com (B.L.)

**Keywords:** indoor air quality, subway station, airborne pollutants, I/O ratio

## Abstract

In this study, the indoor air quality (IAQ) was investigated in a subway station with fully enclosed platform screen doors in Beijing, China. Eight indoor air pollutants, including PM_2.5_, PM_10_, SO_2_ (sulfur dioxide), NO_2_ (nitrogen dioxide), NH_3_ (ammonia), CO (carbon monoxide), CH_2_O (formaldehyde) and TVOC (total volatile organic compound), were measured for six consecutive days in October 2019. The results indicated that the IAQ in the subway station was basically stable at good levels for most times during the whole measurement period. All eight indoor air pollutants were far below their corresponding maximum allowable concentrations, except for the PM_2.5_ concentrations, which occasionally exceeded the concentration limits. The concentrations of indoor air pollutants in the subway station were basically within the corresponding standards. The correlation analyses showed that outdoor air pollutants have important influences on indoor air pollutants. The concentrations of PM_10_, PM_2.5_, SO_2_, NO_2_ and CO in the subway station were positively correlated with their corresponding outdoor concentrations. PM_10_ was statistically significantly correlated with the passenger flow and train frequency, but the other air pollutants were less impacted by the passenger flow and train frequency.

## 1. Introduction

The subway system is convenient and efficient and plays an important role in relieving the burdens of superficial traffic congestion. Meanwhile, the electric power system has been adopted in the subway and has improved the air quality of the city [1,2,3]. However, the internal environment of a subway station platform is relatively confined, which can easily cause various types of trace air pollutants to accumulate, which will lead to potential health risks [4,5]. Epidemiological and toxicological studies show that the concentration of particulate matter, NO_2_ and SO_2_, can affect the cardiovascular, pulmonary functions and respiratory system [6,7,8]. Short-term exposure to PM_2.5_ increases the risk for hospital admission for cardiovascular and respiratory diseases [9]. Long term exposure to PM_2.5_ increases respiratory disease, chronic lung disease, and mortality [10]. Inhalable CH_2_O can exacerbate asthma symptoms and act as a human carcinogen [11,12]. Long-term exposure to TVOC can easily result in childhood leukemia [13]. CO is an inorganic compound that can bind with hemoglobin and reduce the oxygen carrying capacity of red blood cells. More than that, exposure to CO may result in vision loss and diabetes [14,15]. NH_3_ has toxic effects on the central nervous system of the human body, which can lead to behavioral disorders [16]. Consequently, it is of great significance to investigate the indoor air quality (IAQ) of subway stations, to reduce the potential health risks to commuters, via evaluating the concentrations of above-mentioned airborne pollutants.

In recent years, many previous researchers have investigated the IAQ of subway stations in many countries [17,18,19,20,21,22,23,24,25,26,27,28,29,30,31,32,33,34,35,36]. Song et al. [22] reported that the concentration ranges of PM_10_ and PM_2.5_ were 112–159 μg/m^3^ and 52–75 μg/m^3^, respectively, on a Beijing subway platform, and these concentrations were lower than the corresponding outdoor concentrations. Moreover, the authors indicated that the outdoor environment and the service time of the subway had significant effects on the concentrations of airborne particulate matter. The IAQ test results of Martins et al. [26] showed that the concentrations of airborne particulate matter on the platform were approximate 1.3–1.5 times higher than those in the outdoor environment at Barcelona subway station. They confirmed that the concentrations of airborne particulate matter on the platform were mainly correlated with seasonal differences, the design of the station and tunnels, the train frequency, the passenger flows and the change of ventilation system. Park et al. [21] tested Seoul, Korea subway stations, and showed that average concentrations of CH_2_O and TVOC were 15.4 μg/m^3^ and 156.5 μg/m^3^, respectively. Through correlation analysis and comparison, they indicated that CH_2_O and TVOC were weakly related to the depth of subway station and the season. Another study in Seoul, Korea subway station found that the NO_2_ concentrations were significantly lower than the outdoor concentrations. Although a correlation analysis confirmed that the NO_2_ concentration was related to passenger flow and construction year, these factors may not directly affect the NO_2_ concentration [18]. According to Moreno et al. [27], narrow platforms served by single-track tunnels were heavily dependent on the forced tunnel ventilation and cannot rely on the train piston effect alone to reduce platform PM concentrations. In contrast, PM levels in subway stations with spacious double-track tunnels were not greatly affected when the tunnel ventilation was switched off. Simultaneously, their test results for indoor and outdoor concentrations showed that the CO concentrations in the Barcelona metro were very low.

During the daily operation of trains, a certain amount of airborne particulate matter is generated from the friction between the railway and the wheel brake system [26]. In addition, the piston effect produced by the movement of the trains also brings outdoor pollutants into the platform, which affects the air quality of the subway platform [32]. In recent years, screen doors have been installed in many subway platforms. Fully enclosed platforms can separate the platform and the tunnel. This can isolate the heat dissipated by trains from the platform and improve the air quality inside the subway platform [37,38,39,40,41,42,43,44,45]. In this study, a total of eight indoor air pollutants were measured for six consecutive days and analyzed to evaluate the integrated indoor air quality level on a subway platform with fully enclosed platform screen doors. The research could provide a reference for the IAQ of a subway station and its influencing factors.

## 2. Method

### 2.1. Field Study

Eight air pollutants in a subway station platform were measured from 7:00 to 23:00 daily for 6 days, from 22 to 27, October 2019. CPR-KA, an integrated environmental monitor, was used to monitor airborne pollutant concentrations inside a subway station in Beijing, China. Its pump suction rate was 300 mL/min, and its sampling period was 2 min. The measurement range and precision of CRP-KA are shown in Table 1.

The measured subway station was a non-transfer station with fully enclosed platform screen doors. It is an underground station with two tracks in a single tunnel, which adopts a separated island platform design pattern with length and width of 120 m and 14 m, respectively (Figure 1a). The environmental monitor was located at a height of 1.2 m in the middle of the platform, as shown in Figure 1b. The design parameters of the heating, ventilation and air-conditioning (HVAC) system were as follows:

(1) The dry-bulb temperature was 28 °C and the range of relative humidity was 40–70% in the station platform for summer rated conditions.

(2) The total ventilation rate was 5.78 × 10^4^ m^3^/h and the fresh air rate was 1.08 × 10^4^ m^3^/h.

The passenger flow and arrival frequency of train were automatically recorded by the subway control centre. The daily outdoor air pollutant data, including PM_2.5_, PM_10_, CO, NO_2_, SO_2_ and the outdoor atmospheric environment quality index, were retrieved from the website http://beijingair.sinaapp.com/. The data sampling frequency was 1 h.

### 2.2. Data Analysis

Statistical analysis was performed using SPSS 25.00 (Armonk, NY, USA: IBM Corp.) Spearman’s correlation analyses were used to examine the relationships between indoor air pollutants and their factors, including the corresponding outdoor concentrations, the train frequency, and the passenger flow. Differences were considered significant when *p* < 0.05 [46].

In addition, an integrated air quality index (AQI) [47] was adopted to evaluate the indoor air level in the subway station, as shown in Equation (1).
(1)$$\mathrm{AQI}=\sqrt{\frac{{(\max(\frac{c_{1}}{c_{\max1}},\frac{c_{2}}{c_{\max2}},\ldots ,\frac{c_{i}}{c_{\mathrm{maxn}}}))}^{2}+(\frac{1}{n}\sum_{i=1}^{n}\frac{c_{i}}{c_{\mathrm{maxi}}}{)}^{2}}{2}}$$
where *c*_i_ is the concentration of the ith air pollutant, *c*_maxi_ is the maximum permission concentration of *c*_i_, and *n* is the number of measured air pollutants (here *n* = 8).

The integrated AQI can be classified into five levels in consideration of the risks to occupant health, as shown in Table 2 [47].

According to some indoor air quality standards [48,49,50,51], the maximum permissible concentrations of air pollutants are listed in Table 3.

## 3. Results

### 3.1. Passenger Flow and Train Frequency

The passenger flow and train frequency are shown in Figure 2. Day 1 to day 4 represent the weekdays of Tuesday to Friday, and day 5 to day 6 represent the weekend days of Saturday and Sunday. As shown in Figure 2, the train frequency and passenger flow on the weekdays were obviously higher than those on the weekends during the peak hours. The passenger flow peaks in the subway station were at 8:00–9:00 and 18:00–9:00 on weekdays. The average passenger number was 67,126 per hour. The passenger traffic was much busier during the morning peak. There was no clear difference in train frequency and passenger flow during the off-peak hours between weekdays and the weekend.

### 3.2. Air Pollutant Concentrations

Figure 3 and Table 4 illustrate the variations of indoor air pollutant concentrations in the subway station. The variations of indoor NH_3_ concentrations ranged from 0.012 mg/m^3^ to 0.014 mg/m^3^, as shown in Figure 3a. The indoor NH_3_ concentrations were basically stable at a low level, and did not exceed the maximum permissible concentration of 0.2 mg/m^3^. Figure 3b shows that the concentrations of indoor CH_2_O were from 0.008 mg/m^3^ to 0.079 mg/m^3^. Most of the concentrations were below 0.08 mg/m^3^ and did not exceed the maximum permissible concentration of 0.12 mg/m^3^. Figure 3c depicts the concentrations of indoor TVOC remaining in the range between 0.374 mg/m^3^ and 0.423 mg/m^3^. The TVOC concentrations kept quite consistent during the test period and did not exceed the maximum permissible concentration of 0.6 mg/m^3^.

The indoor NO_2_ concentrations changed notably with time from 0.006 mg/m^3^ to 0.127 mg/m^3^, as shown in Figure 3d, but they remained below the maximum permissible concentration of 0.24 mg/m^3^. The indoor NO_2_ concentrations increased markedly from 17:00 and reached their peaks at 20:00–21:00, except for on day 3.

In Figure 3e, the indoor SO_2_ concentrations fluctuated in the range between 0.001 mg/m^3^ to 0.007 mg/m^3^ and remained below the maximum permissible concentration of 0.5 mg/m^3^. The indoor SO_2_ concentrations rose from 11:00 to their peak values at approximately 16:00, and then decreased. The daily trends were similar throughout the whole test period.

Figure 3f shows that the variations of indoor CO concentrations were from 0.046 mg/m^3^ to 0.111 mg/m^3^. These were below the maximum permissible concentrations during the test period. From day 1 to day 3, the indoor CO concentrations fluctuated with time. However, the peak values appeared at different times. From day 4 to day 6, the indoor CO concentrations did not obviously fluctuate with time. Hence, the indoor CO concentrations were less impacted by the changes of train frequency and passenger flow.

The concentration ranges of indoor PM_2.5_ and PM_10_ were from 0.006 mg/m^3^ to 0.196 mg/m^3^ and from 0.008 mg/m^3^ to 0.237 mg/m^3^, respectively, as shown in Figure 3g,h. The indoor PM_10_ concentrations did not exceed the maximum permissible concentration of 0.25 mg/m^3^. The average indoor PM_2.5_ concentrations also remained below the maximum permissible concentration of 75 μg/m^3^, except for on day 2. Except for that on day 3, the peaks of indoor PM concentrations occurred between 19:00 and 21:00. Although their concentrations fluctuated with time, their change trends were different from the trends of passenger flow and train frequency.

### 3.3. AQI

Figure 4 shows variations of indoor AQI during the days of investigation. Most of the days, except for day 2, showed values below 0.5 and remained at a good level. The change range of AQI on day 2 was approximately 0.6–0.7. The AQI level during day 2 was at an acceptable level which was affected by the serious outdoor air pollution.

## 4. Discussion

### 4.1. Variations of Indoor Air Pollutants

The NH_3_ was mostly generated indoors, such as from the toilets on the platform [52]. The indoor NH_3_ has been well diluted by the HVAC system to maintain a low level far below the concentration limit. The indoor CH_2_O mainly accumulated from the emissions of building materials, furniture and various adhesive coatings [53]. The change of indoor CH_2_O concentrations could be related to indoor temperature. Higher indoor temperature can be helpful for the release of more CH_2_O from the building finishing materials [54]. This might explain the increase of CH_2_O concentrations which occurred at the morning or evening peaks. The TVOC concentrations remained stable during the test period, because the TVOC mostly came from the building material emissions [55]. In sum, the concentrations of NH_3_ and TVOC (including CH_2_O) were mostly generated indoors and kept relatively stable during the test period by the ventilation of the HVAC system.

Figure 5 shows the variations of outdoor air pollutant concentrations during the measurement. It can be seen that the daily variations of indoor NO_2_, SO_2_, CO, PM_2.5_ and PM_10_ concentrations were quite consistent with the corresponding variations of outdoor concentrations. The indoor NO_2_, SO_2_ and CO mainly came from the exhaust of motor vehicles introduced through the HVAC system and subway entrances [56]. Similarly, a large portion of indoor PM_10_ and PM _2.5_ came from the road re-suspension dust and vehicular emissions [57], which were also brought in by the ventilation of HVAC system or directly through the entrances. Meanwhile, most of vehicle exhausts were found to be composed of fine aerosol lower than 2.5 μm. Thus, the daily change trends of indoor PM_2.5_ and PM_10_ were well correlated (Figure 3g,h), which was consistent with the findings of Park et al. [58] Consequently, the indoor NO_2_, SO_2_, CO, PM_2.5_, PM_10_ concentrations basically fluctuated with their corresponding outdoor concentrations. Meanwhile, their indoor concentrations were basically lower than the outdoor concentrations due to the filtration and dilution by the ventilation of HVAC system.

In general, the peaks of indoor concentrations of these five pollutants mainly occurred during the morning or evening rush hours. Therefore, highly congested traffic situations during the peak hours may exacerbate the IAQ of subway station under the ground vehicle road. There were bus stops located next to the subway station entrance so that passengers connect conveniently, which could also contribute to the variations of the pollutants.

### 4.2. Comparison with Previous Studies

Table 5 shows the indoor air pollutant concentrations from other references. As shown in Table 5, the studies used for the comparison were mostly conducted in the summer and transitional season, with HVAC systems in operation. In our study, the measurement campaign was performed in late October (transitional season), when the weather in Beijing was mild, but the HVAC system of the subway station was still operating in cooling mode due to the high passenger flow. The average passenger numbers given in the few studies were also comparable to the average passenger flow of the subway station investigated in our study. Most of the previous studies shown in Table 5 have investigated multiple subway stations, but the stations size and ventilation system parameters could not be compared, due to a lack of relevant information in these studies.

The average indoor NH_3_ concentration of 0.012 mg/m^3^ in our study was relatively low, compared with the NH_3_ concentration given in the references [59]. The average indoor CH_2_O and TVOC concentrations were 0.035 mg/m^3^ and 0.405 mg/m^3^, which were much higher than the concentrations on the Seoul subway platforms [21] and the Taipei subway platform [60]. They also indicated that the indoor TVOC (including CH_2_O) concentrations had no correlation with the number of passengers, but had a weak correlation with the depth of the platform. This support our findings that the indoor TVOC (including CH_2_O) concentrations could be primarily attributed to the emissions of interior building materials. The higher TVOC concentrations measured in our study were probably caused by the emissions of detrimental decoration materials.

The average NO_2_ concentration in our study was slightly lower than the average concentration on the Seoul subway platforms [18]. The I/O ratios of NO_2_ in our study were also quite similar to the I/O ratios of 0.59–0.74, as indicated in the reference [18]. The higher outdoor concentrations of NO_2_ could be attributed to the diesel exhaust fumes from motor vehicles on the roads in urban areas.

The average SO_2_ concentration in our study was 0.003 mg/m^3^, which was much lower than the concentrations reported in the Guangzhou subway stations [61]. The average indoor CO concentration of 0.059 mg/m^3^ was much lower the average concentration reported in the Taipei subway stations [60], but quite comparable with the average concentration in the Nanjing subway stations [62]. There was no indoor source for CO and SO_2_ in the subway station, therefore the indoor CO and SO_2_ basically came from the contaminated ambient air being brought down from street level. The relatively low indoor CO and SO_2_ concentrations in the Beijing subway station indicated a good ventilation performance by the HVAC system.

The average PM_10_ concentration of 0.061 mg/m^3^ was lower than the concentrations reported in the subway stations in Taipei [60], Nanjing [62] and Seoul [63]. The average PM_2.5_ concentration was 0.048 mg/m^3^, which was also lower than the concentrations reported in the references [60] and [58]. The lower PM concentrations observed in our study could be attributed to both the platform screen doors and the good ventilation performance of the HVAC system. There is a certain amount of PM generated from the train operation [26]. Several researchers have indicated that the fully enclosed platform screen doors could help prevent the PM generated by the train operation from entering the platform [26,41]. In addition, the screen doors could also prevent a portion of outdoor air pollutants from entering the platform through the piston wind in the tunnel [42]. Nevertheless, the indoor space of the station would be decreased by installing the fully enclosed screen doors, which might result in a slight increase of other indoor air pollutant concentrations.

It is worth noting that the majority of indoor PM was still introduced from outdoors through the HVAC system and station entrances, which could not be prevented by screen doors. As shown in Table 5, high PM_10_ concentrations were observed in the Nanjing subway stations [62], which could be attributed to the ventilation method they used in the transitional season. During the time of sampling, they used natural ventilation systems instead of HVAC systems, which no doubt fully reduced both the ventilation rates and filtration efficiency. Similarly, the high PM_10_ concentrations reported in the Seoul subway stations were also caused by insufficient air circulation and improper ventilation [56]. In their study, the PM_10_ concentrations on platforms were even obviously higher than those outdoors, because the ventilation was insufficient to remove the accumulated particles brought in from outdoors. Therefore, the proper operation of the HVAC system was also crucial to control the concentrations of indoor PM and other pollutants to maintain them at acceptable levels.

### 4.3. I/O Ratios

Figure 6 shows the indoor and outdoor (I/O) ratios of PM_2.5_, PM_10_, SO_2_, NO_2_ and CO concentrations. The indoor PM concentrations in the subway station fluctuated with the variations of the corresponding outdoor concentrations, as shown in Figure 6a,b. It was reported that some particles would be generated in the subway, due to the friction between the track and the wheel [26]. In addition, when the passenger flows were large, the airborne particulate matter from the floor would be re-suspended, due to the passenger movement around the subway platform [58]. Hence, increased passenger flow may cause an increase in the particle concentration in the subway platform. The I/O ratios of PM_2.5_ and PM_10_ were within the ranges of 0.77–2.34 and 0.57–1.58, respectively. During most of that time, the indoor PM concentrations were smaller than the outdoor concentrations, which indicated that the fully enclosed platform screen doors could prevent the generation of pollutants from the train running [64]. Thus, the PM in the subway station mainly came from the outdoor environment through the HVAC system and the entrances. On days 3 and 4, the indoor PM concentrations were higher than the outdoor concentrations, which might have been affected by the concentrations of the previous day. The air conditioning system was switched off after the last train every day, possibly resulting in the accumulation of indoor air pollutants on the platform. Therefore, the average indoor air pollutants’ concentrations could be affected by the high concentration in the previous day, such as the day 2 in this study.

SO_2_ is the combustion product of coal or oil, and is mainly associated with industrial sources [65]. There was no SO_2_ production source in the subway station. Indoor SO_2_ was mainly affected by the outdoor SO_2_ through the ventilation. As shown in Figure 6c, the indoor SO_2_ concentrations were mainly consistent with the outdoor SO_2_ concentration. The I/O ratios of SO_2_ were in the range between 0.44 to 2.15. Similar to the indoor PM concentrations, the indoor SO_2_ concentrations were also higher than the outdoor concentrations on days 3 and 4.

The indoor NO_2_ concentrations were lower than the outdoor NO_2_ concentrations, and the I/O ratios were from 0.45 to 0.81, as shown in Figure 6d. It is generally believed that the NO_2_ is mainly caused by the emission of outdoor road diesel vehicles [66]. The indoor NO_2_ concentrations were influenced by the outdoor NO_2_ concentrations.

The indoor CO concentrations were much lower than the outdoor CO concentrations, and the I/O ratios were from 0.06 to 0.12, as shown in Figure 6e. CO is produced by incomplete combustion. The indoor CO concentrations are at relatively low levels, because there is no chemical combustion or smoking in the subway station. Hence, the indoor CO might come from the traffic-contaminated air from outdoors [29].

### 4.4. Influencing Factors

Table 6 lists the correlation analysis between indoor air pollutants and their influencing factors, including the corresponding outdoor concentrations, train frequency and passenger flow. Furthermore, the correlations between the indoor AQI and the outdoor atmospheric environment quality index, train frequency and passenger flow were also analyzed.

The results showed that the indoor PM_10_ concentrations were statistically significantly correlated with the outdoor PM_10_ concentration (*r* = 0.858, *p*< 0.01), the passenger flow (*r* = 0.201, *p* < 0.05) and the train frequency (*r* = 0.209, *p* < 0.05). Other air pollutant concentrations were strongly correlated with their corresponding outdoor concentrations, but less impacted by the passenger flows and train frequency. The AQI also had a significant correlation with the outdoor atmospheric environment quality index (*r* = 0.649, *p* < 0.01). Hence, the outdoor air pollutants had significant contributions to the indoor concentrations through the HVAC system. The variations of indoor concentrations of SO_2_, CO, NO_2_, PM_10_ and PM_2.5_ were most likely related to their corresponding outdoor concentrations.

According to the correlation analysis, the indoor PM_2.5_ concentrations and gaseous pollutants were only correlated to the outdoor environment. In contrast, the indoor PM_10_ concentrations were not only affected by the outdoor environment, but also related to the passenger flow and the train frequency. Martins et al. [26] indicated that the PM_2.5_ concentrations in subway platforms with screen doors were lower than those in open subway stations. Therefore, the fully enclosed platform screen doors can better prevent the fine particles produced by the trains from moving to the platform.

## 5. Conclusions

In this study, eight airborne pollutants in a subway station with fully enclosed screen doors were consecutively measured for six days in Beijing, China. The IAQ performance of the station has been evaluated comprehensively, and compared with previous studies. The potential influencing factors of IAQ were also discussed. Future studies were recommended to investigate more subway stations with different station sizes, passenger flows, platform types and ventilation systems, meanwhile covering more outdoor climate conditions. The main conclusions of this study are summarized as follows:

(1) The concentrations of indoor air pollutants on the subway platform were basically within the corresponding standards. The AQI were at good and acceptable levels during the whole measurement.

(2) The concentrations of NH_3_ and TVOC (including CH_2_O) were kept relatively stable during the test period, because they were mostly generated from indoor emission sources and were well diluted by the ventilation of HVAC system.

(3) The concentrations of indoor PM_10_, PM_2.5_, SO_2_, NO_2_ and CO were positively correlated with their corresponding outdoor concentrations. The daily variations of these indoor air pollutant concentrations were also influenced by the corresponding variations of outdoor concentrations to a large extent. The indoor concentrations were generally lower than the outdoor concentrations, due to the filtration and dilution by the HVAC system.

(4) Except for the indoor PM_10_, the other indoor pollutants and the overall air quality had no statistically significant correlation with the passenger flow and the train frequency. Therefore, the fully enclosed platform screen doors can effectively prevent the fine particles produced by the train operation from moving into the platform area. However, it is worth noting that the indoor pollutants were still mostly introduced from outdoors through the HVAC system and subway entrances, as indicated by the correlation analyses, which could not be prevented by screen doors. The proper operation of HVAC system was also crucial to control the indoor pollutant concentrations at acceptable levels.

## Figures and Tables

**Figure 1 ijerph-17-05213-f001:**
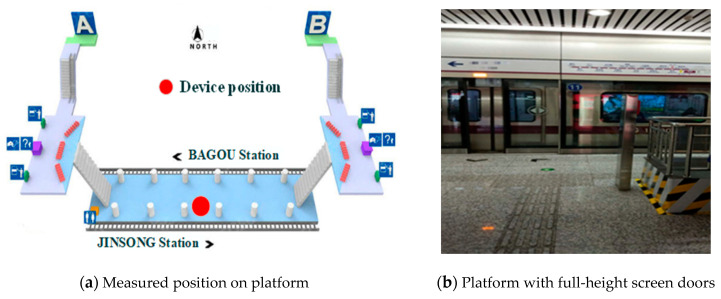
Measured position and platform. (**a**) Measured position on platform, (**b**) Platform with full-height screen doors.

**Figure 2 ijerph-17-05213-f002:**
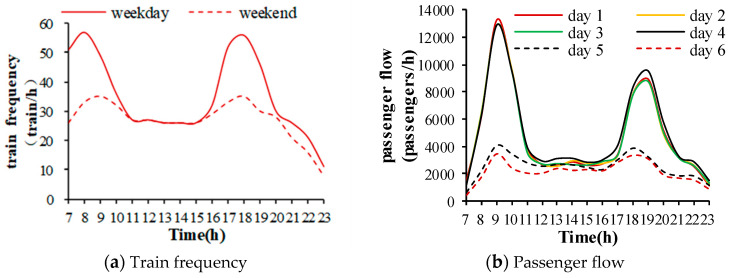
Variations of daily train frequency and passenger flow. (**a**) Train frequency. (**b**) Passenger flow.

**Figure 3 ijerph-17-05213-f003:**
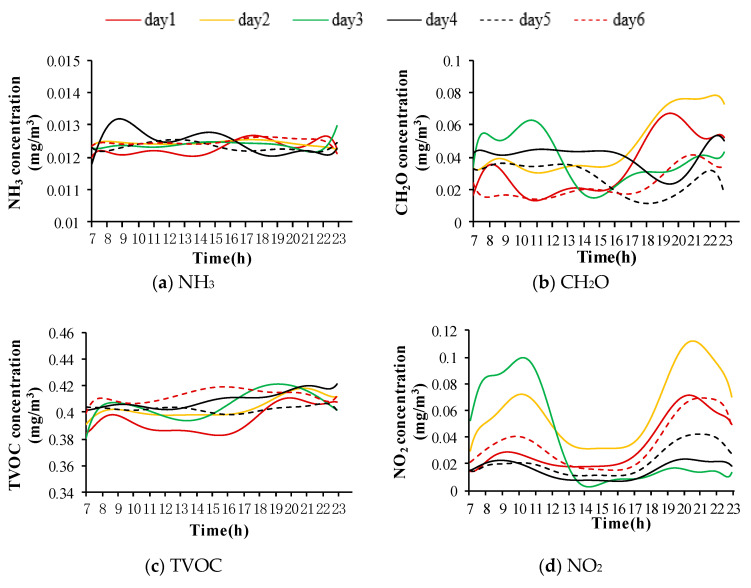
Variations of indoor air pollutant concentrations in the subway station. (**a**) NH_3_, (**b**) CH_2_O, (**c**) TVOC, (**d**) NO_2_, (**e**) SO_2_, (**f**) CO, (**g**) PM_2.5_, (**h**) PM_10_.

**Figure 4 ijerph-17-05213-f004:**
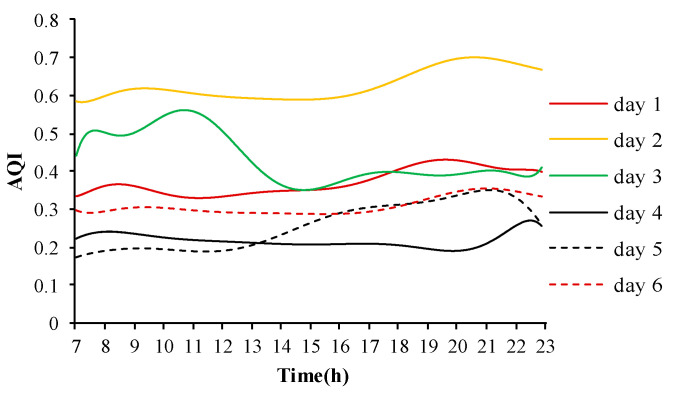
Variations of indoor AQI in the subway station.

**Figure 5 ijerph-17-05213-f005:**
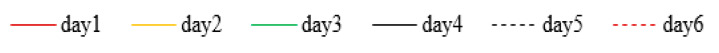
Variations of outdoor air pollutant concentrations during the measurement. (**a**) NO_2_, (**b**) SO_2_, (**c**) CO, (**d**) PM_2.5_, (**e**) PM_10_.

**Figure 6 ijerph-17-05213-f006:**
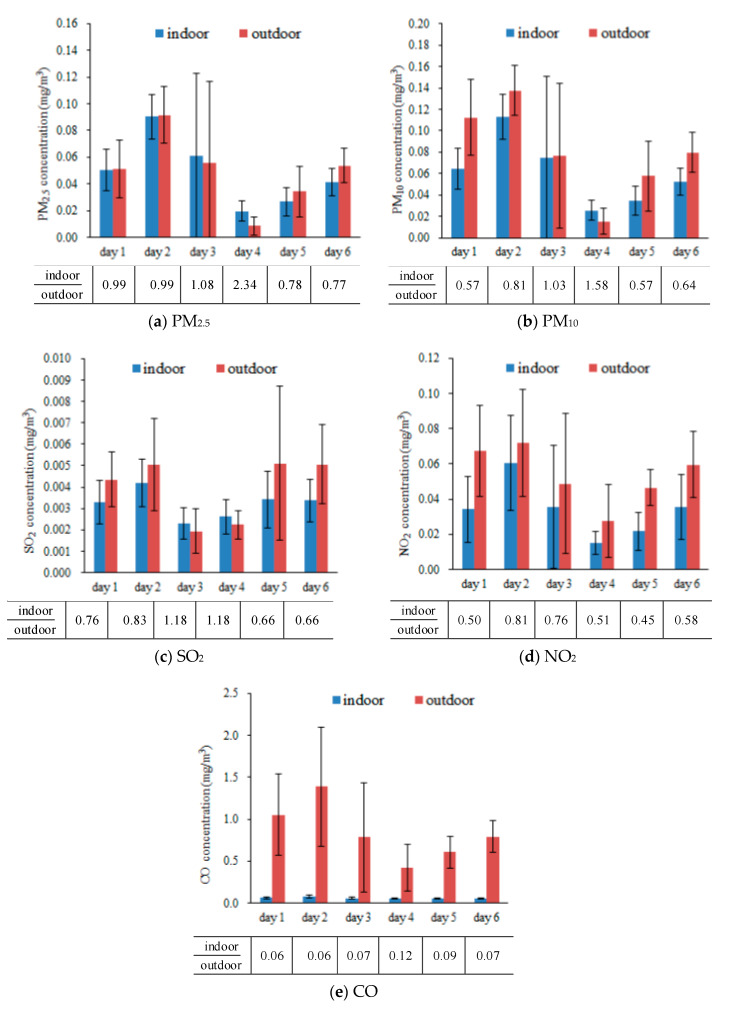
Comparison between the indoor and outdoor air pollutant concentrations. (**a**) PM_2.5_, (**b**) PM_10_, (**c**) SO_2_, (**d**) NO_2_, (**e**) CO.

**Table 1 ijerph-17-05213-t001:** Measurement range and precision of CPR-KA.

Indoor Air Pollutants	Measurement Range	Precision
NH_3_	0–30 ppm	1 ppb
TVOC	0–10 ppm	1 ppb
CO	0–50 ppm	1 ppb
CH_2_O	0–10 ppm	1 ppb
NO_2_	0–2 ppm	0.1 ppb
SO_2_	0–2 ppm	0.1 ppb
PM_10_	0–0.5 mg/m^3^	0.001 mg/m^3^
PM_2.5_	0–0.5 mg/m^3^	0.001 mg/m^3^

**Table 2 ijerph-17-05213-t002:** Classification standard of integrated article air quality index (AQI).

Integrated AQI	Air Level	Implication
0–0.5	Good	Air quality is satisfactory.
0.5–1.0	Acceptable	Air quality is acceptable. There may be some risks for unusually sensitive groups.
1.0–1.5	Slight	One air pollutant exceeds its limit value. There are potential health risks for the susceptive groups.
1.5–2.0	Moderate	Two or three air pollutants exceed their limit values. There are health risks.
>2.0	Heavy	More than three air pollutants exceed their limit values. There are serious health risks.

**Table 3 ijerph-17-05213-t003:** Maximum permissible concentrations of indoor air pollutants.

Air Pollutants	Concentration Limit	References	Time-Average
CO	10 mg/m^3^	[48,49,50]	1 h average
CH_2_O	0.12 mg/m^3^	[48]	*n*/a
TVOC	0.6 mg/m^3^	[50]	8 h average
SO_2_	0.5 mg/m^3^	[50]	1 h average
NH_3_	0.2 mg/m^3^	[50]	1 h average
NO_2_	0.24 mg/m^3^	[50]	1 h average
PM_10_	0.25 mg/m^3^	[48,49]	*n*/a
PM_2.5_	75 μg/m^3^	[51]	24 h average

**Table 4 ijerph-17-05213-t004:** Indoor air pollutant concentrations in the subway station.

Pollutants	Min (Mg/M^3^)	Max (Mg/M^3^)	Mean ± SD (Mg/M^3^)	Maximum Permissible Concentration (Mg/M^3^)
NH_3_	0.012	0.014	0.012 ± 0.0004	0.200
CH_2_O	0.008	0.079	0.035 ± 0.0161	0.120
TVOC	0.374	0.423	0.405 ± 0.0092	0.600
NO_2_	0.006	0.127	0.034 ± 0.026	0.240
SO_2_	0.001	0.007	0.003 ± 0.0012	0.500
CO	0.046	0.111	0.059 ± 0.0144	10.000
PM_10_	0.008	0.237	0.061 ± 0.044	0.250
PM_2.5_	0.006	0.196	0.048 ± 0.036	0.075

**Table 5 ijerph-17-05213-t005:** Indoor air pollutants concentrations measured in subway stations in previous studies.

Pollutant	Average Concentration	City	Reference	Platform Type	Season	Average Passenger Per Hour
NH_3_	119.63 ± 3.06 µg/m^3^	Kunming	[59]	Fully enclosed platform	*n*/a	*n*/a
CH_2_O	15.4 ± 7.2 µg/m^3^	Seoul	[21]	Fully enclosed platform	Summer	45,115
CH_2_O	0.017 ± 0.016 mg/m^3^	Taipei	[60]	Both fully and semi enclosed platforms	Summer	*n*/a
TVOC	0.064 ± 0.035 ppm	Taipei	[60]	Both fully and semi enclosed platforms	Summer	*n*/a
TVOC	156.5 ± 78.2 µg/m^3^	Seoul	[21]	Fully enclosed platform	Summer	45,115
NO_2_	0.053 ± 0.008 mg/m^3^	Seoul	[18]	Fully enclosed platform	Summer	37,908
SO_2_	0.13 ± 0.01 mg/m^3^	Guangzhou	[61]	Fully enclosed platform	Summer	*n*/a
CO	2.825 ± 0.69 mg/m^3^	Taipei	[60]	Both fully and semi enclosed platforms	Summer	*n*/a
CO	0.3 ± 0.2 mg/m^3^	Nanjing	[62]	Fully enclosed platform	Transitional season	*n*/a
PM_10_	0.185 ± 0.128 mg/m^3^	Nanjing	[62]	Fully enclosed platform	Transitional season	*n*/a
PM_10_	90.7 ± 9.9 µg/m^3^	Seoul	[63]	Fully enclosed platform	Summer	57,251
PM_10_	80.9 ± 34.9 µg/m^3^	Taipei	[60]	Both fully and semi enclosed platforms	Summer	*n*/a
PM_2.5_	105.4 ± 14.4 µg/m^3^	Seoul	[58]	*n*/a	Winter	*n*/a
PM_2.5_	56.2 ± 33.1 µg/m^3^	Taipei	[60]	Both fully and semi enclosed platforms	Summer	*n*/a

**Table 6 ijerph-17-05213-t006:** Correlation coefficients between indoor pollutants and influencing factors.

Factors	Indoor Air Pollutants and AQI								
	PM_2.5_	PM_10_	SO_2_	NO_2_	CO	NH_3_	TVOC	CH_2_O	AQI
Corresponding outdoor values	0.951 **	0.858 **	0.732 **	0.868 **	0.915 **	n/a	n/a	n/a	0.649 **
Passenger flows	0.190	0.201 *	–0.129	0.125	0.149	0.128	−0.012	0.184	0.164
Train frequency	0.198	0.209 *	−0.136	0.143	0.170	0.098	−0.068	0.199	0.164

** *p* < 0.01, * *p* < 0.05.

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
