# Peer review of "Experimental Investigation of Air Quality in a Subway Station with Fully Enclosed Platform Screen Doors"

_ijerph, 2020, doi:10.3390/ijerph17145213_

Round 1
Reviewer 1 Report
I believe that the content from this study inside the scope of IJERPH, but the manuscript must be improved for its publication. The write-up of the manuscript looks more as a “technical report” that a “scientific paper”. I recommend that authors rewrite the paper and submit again for a new evaluation.
- General Comments
- As I said previously, this manuscript looks more as a “technical report”, and this can be seen when the authors write “In Fig. x…” All results are written in the same form, and this is not recommendable for scientific paper.
- The results only describe the figures or tables instead of to employ as support for give the reader information. This must be improved.
- The discussion is very poor, the authors only mentioned “this result was different from this other” but they don’t give more information about the reasons because the results differ.
- The authors used many abbreviations (for instance, CH2O or TVOC) but they don’t say the mean of every one. The only one exception was indoor air quality (IAQ) in the abstract.
- Tables title have different format and are not as journal recommended.
- Must be employed Figure not Fig. in the figures title.
- References format must be revised and to homogenize them according to journal specifications.
- Introduction
- Lines 29-31: The sentences must be referenced.
- Lines 40-41: For the first time IAQ is mentioned, but non-information is given about this index, and its importance for being included in this study.
- Lines 43-45: The reference of Song et al. is collocated at the end of sentence, whilst that in the other sentences is after the authors. See line 47, Martins et al. [23].
- Lines 46-48: The author said: “the test results of Martins et al. [23]…” What test did Martins et al. make?
- Line 47: “Concentration of particles” referred to particulate material? To be more specific with the information.
- Line 49: “Particles concentration”. See previous comment.
- Lines 57-58: “According to the study by T. Moreno” change by: “According to Moreno”
- Lines 57-62: The authors mentioned study of Moreno et al. [24], but they compared single-track tunnel with forced ventilation systems and double-track tunnel with closed ventilation systems. I understood that Moreno et al. [24] evaluated both types of tunnels (single and double) with forced ventilation and with forced ventilation turned off. So, what did occur with single-track tunnel with closed (forced ventilation turned off) ventilation systems and double-track tunnel with forced ventilation systems? Thus, the information could be compared. Moreover, the authors must avoid employing concepts that can confuse the reader.
- Lines 60-62: The sentences must be referenced.
- Line 63: Same comment that line 49.
- Lines 66-68: The title have the words “fully enclosed platform screen doors”, but only at the end of introduction the authors mentioned them. How this kind of doors could affect the air conditions? The authors give two references [35-36] where it is indicated that improve air quality, but more information in introduction about this must be provided.
- In general, I believe that other results must not be included in the introduction (i.e. Park et al. [17]). This information must be employed in discussion and there to develop the idea in deep. They can use the same references for contextualizing to reader about the importance of topics or for demonstrating that it has been employed in other places.
- Method
- 1 is not a contribution for this study, if only show us the equipment.
- A station description is required for improving the comprehension of the results.
- In results there is information about passenger flow and train frequency, but in method there is not information about as this was obtained.
- In data analysis, outdoor pollutions are mentioned, but they did not mention from where this information was obtained.
- Results
- This section must be rewritten completely as scientific paper.
- The measurement units must be the same. The authors worked with ppb and mg m-3 indistinctly, but for readers is better to employ only one (ppb or mg m-3).
- The authors only must explain the results obtained, without citing other authors or discussing. All section must be revised
- Line 127-129. The authors said that in Day 3 the NO2 reached a peak, but in Figure 3d was in the Day 2 when peak was reached.
- Line 142. Table 6 is indicated, but recently Table 3 was mentioned. Tables and figures must be written in correlative order according they are mentioned in the text. Anyway, information in lines 141-142 correspond to discussion section.
- Lines 144-147. The authors mentioned that PM10 didn’t exceed the permissible concentration, except for Day 2, indoor PM5. This must be revised and compared PM10 with PM10 and PM2.5 with PM2.5.
- In Figure 3. There are graphics with very erratic behavior. For instance, CH2O showed a peak at 8:00-9:00 but not at 18:00-19:00 for Day 2 (weekday), and this was also registered for NO2, CO, PM5 and PM10. Even, the concentration was least that at weekend days at 18:00-19:00. In change, days 1 and 3 showed a contrary situation, registered peak at 18:00-19:00. This must be developed in discussion. Why are those differences registered? All graphics must be revised and deeply discussed.
- In Figure 3, the authors mentioned that concentration of some pollutants varied between days. How did they determine that variation? I believe that a statistical analysis between days for each pollutant could collaborate with that information. Especially because in methods is mentioned that each 2 min a sample is taken.
- In Figure 3, How could the authors explain that SO2 increase in weekend days?
- Discussion
- This section must be rewritten completely as scientific paper.
- Lines174-187. The authors compared their results with other references, but they only mentioned there is a different concentration in this study in comparation with other. However, I would like to know if the station conditions of those studies are different with this study. The references made their studies under the same ventilation systems or the station size or any information that allow to identify why that differences were registered.
- In Table 5, some references indicated the season (summer or winter), but this study was realized in autumn. I don’t know if these results can be compared. Conversely, I don’t know the climatic conditions on the study place, but this must be clarified for doing a better discussion.
- In Figure 5, there are huge error bars for indoor and outdoor air pollutants. How can you explain this?
- In Figure 5, where did the authors get the outdoor air pollutant information?
- In Figure 5, there are five of eight pollutants, what did to the other three pollutants occur?
- Lines 202-203. The sentences must be referenced. Because this is not obtained from this study.
- Lines 209-210; 215-216. The authors only mentioned that higher concentration of indoor pollutants might be by the previous days, but they don’t give more information about this. How did they determine this? Is there another alternative or suppose that could explain this behavior?
- Lines 223-224. The authors indicated that low CO concentrations must be because there is not combustion and smoking inside the station. This could be true, but why in this case outdoor pollutant did not affect inside concentration? Similarly, with SO2, where there is not emission inside the station, but the only one reason the indoor concentration was outdoor pollutants.
- Lines 236-237. Say r=0.201, p<0.01 and r=0.209, p<0.01, but according to Table 6 the significant was 0.05 in these cases.
- Conclusions
- This section must be rewritten completely as scientific paper.
- Lines 244-247. The authors mentioned that indoor air pollution in Beijing subway station were relatively low in comparison with literature. I believe that they cannot concluded this without additional information about the station conditions, both from Beijing or another place.
- Lines 248-250. I don’t know if it possible to get to this conclusion, due to the erratic behavior of the pollutants. Especially, because in Figure 3 there are some pollutants peaks related with peak of train frequency and passenger flow in morning peak but none in evening peak.
- Lines 251-254. Even though this conclusion has related with the results obtained, I believe that authors contradict themselves when there is a positive correlation between PM10 with train frequency and flow passenger.
- Lines 256-257. I can see a positive correlation between PM10 with train frequency and flow passenger, but for me a correlation of 0.201 or 0.209 is not strong correlation but a weak correlation. This correlation can be significant, but this is not the same to strong.
Reviewer 2 Report
This study analyzes the indoor air quality of a subway station by measuring 8 indoor air pollutions. The article is interesting to the International Journal of Environmental Research and Public Health. However, several aspects have been detected that should be addressed by the authors:
The novelty and originality of the study should be better explained. The contributions of the current study to that shown in other studies are not clear.
A flowchart should be included that summarizes the research steps.
More details of the analyzed station should be given. Plans or 3D views of the station would help readers. The measuring equipment should be located in these figures.
I don't think writing the results using bullet points is the most appropriate. Authors should write the results using paragraphs.
Table 5. Reference [44].
Table 6. It is not clear how the correlation coefficient and p-value are obtained in each variable. Authors should better explain this aspect.
The figures should be better explained. In the current format, many figures are not sufficiently analyzed.
The format of the conclusions is not adequate. The authors should explain the main conclusions of the research and not list them.
There are English problems; I suggest that some college whose native language is English revise the manuscript.
Reviewer 3 Report
The manuscript submitted by Pang et al. concerns air quality monitoring in a subway station in Beijing. Monitoring results are reported for the concentrations of PM2.5, PM10, SO2, NO2, NH3, CO, CH2O and TVOC. Measurement results are complemented by outdoor concentrations and a respective correlation analysis. Further correlation analyses are done for train frequency and passenger flow.
The scope of the article combines scientific interests as well as public interests. The article fits well in the focus of the journal and deserves publishing if the following concerns can be addressed:
General comments:
- Proof reading by a native speaker would be beneficial.
- It is necessary to provide detailed information on the outdoor measurements. It is mandatory to provide technical data on measurement methods: (i) measurement principles, (ii) precision, (iii) accuracy, (iv) time resolution of reported data (e.g. daily mean,…).
- Moreover, the authors must describe how the correlation analyses were done. Currently, there is no statement on the analysis method.
- The authors should clearly specify the concept “air quality level” or “air quality index” and provide a concise definition (e.g. World Health Organization, 2005. WHO Air Quality Guidelines for Particulate Matter, Ozone, Nitrogen Dioxide and Sulfur, Global Update, Summary of Risk Assessment; see also: World Health Organization, 2002. In: Murray, C.J.L., et al. (Eds.), Summary Measures of Population Health: Concepts, Ethics, Measurement and Applications, WHO; see also: Atmospheric Environment, 2017, 166, 570-572, DOI:10.1016/j.atmosenv.2017.06.051).
Specific comments:
- Check if the term “pollution” should be replaced by “pollutant” throughout the whole text.
- Check if the term “Day” should be replaced by “day” throughout the whole text (avoid unnecessary capitalization).
- Abstract, line 20: replace “strong” by “strongly”.
- Introduction, line 50: replace “density of passengers” by “passenger flows”.
- Introduction, line 58: replace “T. Moreno et al.” by “Moreno et al.”.
- Method, line 76: replace “VOC” by “TVOC”.
- Method, table 1: Differentiate between accuracy and precision and add a new column for the precision of the measurements.
- Method, line 84: Provide a reference for the employed software and describe how the correlation analyses were done. Currently, there is no statement on the analysis method.
- Method, table 2: The so-called “air quality indices” vary greatly in the classification of the health risk, depending on the reference values. Please consider a corresponding statement. Provide a reference for table 2.
- Method, table 3: The authors should consider to make a reference to WHO concentration limits.
- Results, line 120: replace “mainly” by “most likely”.
- Results, line 124: replace “mainly” by “most likely”.
- Discussion, lines 177-178: Incude the missing hyperlink for the reference.
- Discussion, table 5: Add the results of this study in column 2 to facilitate a direct comparison.
- Discussion, table 5: Add a statement in column 4, if these studies also deal with fully enclosed subway stations.
- Discussion, line 90: Provide a statement on how the outdoor data were obtained.
- Discussion, figure 5: Think about scatter plots to analyze the I/O correlation on a daily basis.
- Discussion, line 229: Please describe how the correlation analyses were done.
- Discussion, line 235: replace “strong” by “strongly”.
- Conclusion, lines 248-249: Please clarify this statement.
- Conclusion, line 256: replace “strong” by “strongly”.
Round 2
Reviewer 1 Report
General comments
I believe that authors did not really make substantially changes in the manuscript, especially in the discussion section. For instance, the comment: “Lines174-187. The authors compared their results with other references, but they only mentioned there is a different concentration in this study in comparation with other. However, I would like to know if the station conditions of those studies are different with this study. The references made their studies under the same ventilation systems or the station size or any information that allow to identify why that differences were registered”. The authors answer was: “The comparison has been enriched by further considering the effect of screen doors Unfortunately, the ventilation system and station size information given in previous studies was inadequate for a solid comparison”. But, in the manuscript only one sentences was added (lines: 195-198), and it have little scientific support. On the other hand, I can see in Table 5 that PM10 concentrations were very different Seoul [59] and Nanjing [58] but both have fully enclosed platform. So, how can the authors declare that fully enclosed platform could effectively prevent the airborne particulate matter?
They only considered the simplest suggestions, but deeply changes did not include.
Even, Reviewer 2 requested the authors that figures must be better explained. I believe that this is a requirement yet, especially with Figure 3. When I read the manuscript, I could not identify “the improvement in the interpretation of the data shown in these figures” according to the authors answered to Reviewer 2.
Specific comments
Lines 43-44. Previously I requested to authors that they give us information about IAQ. What is IAQ mean? And Why is their inclusion important?
Lines 83-84. The authors give some station characteristics, but I believe that more information about the station is necessary. For instance: They must be indicated if the station is single or double tunnel, because as they mentioned this can have effect over pollutant concentrations (Moreno et al.), or station depth (Park et al.)
Fig. 1a. This is a very useful information, but size and quality must be improved.
Line 126: The sentences: “There were two toilets in the subway, which resulted in a certain amount of NH3” must be translated to discussion because this an assumption, moreover it must be referenced.
Line 129: What is the HVAC mean? I can assume that correspond to air conditioning system.
Lines 137-138. Discussion
Line 158: Discussion (Park et al.)
Figure 3b. In the previous review, I requested to the authors explanation about the erratic behavior of some pollutants, for instance, CH2O showed a peak between 07:00 and 11:00 on day 3, and very different behavior on day 2 where the peak was registered between 18:00 to 22:00. The authors only answer in function of correlation between some pollutants (NO2, CO, PM2.5, PM10 and SO2) between outdoor and indoor concentration, but they don’t give explications about CH2O. This information must be added to discussion.
Figure 3. The authors indicated or justified the NO2, CO, PM2.5, PM10 and SO2 indoor concentration according to daily outdoor concentration, but they did not explain why some pollutants have a difference behavior between hours. For instance, why CO showed a peak early morning (08:00-11:00) on day 3, while that on day 2 the peak was registered at night (18:00-22:00), or How would the authors explain it that NO2 reached almost 0 ppb between 14:00 to 15:00 hours on day 3? Correlation analysis made by the authors do not answer this, and they neither answer them in the previous review. This information must be added to discussion.
Figure 3. The authors also did not answer about the peak of SO2 at weekend days. They said that SO2 is produced by is the combustion product of coal or oil and is mainly associated with industrial sources, in their review answers. I would expect that authors develop will develop deeply the hourly variation for the pollutants because they have a very fluctuating behavior. This information must be added to discussion.
Lines 183-194. I keep my doubts with this discussion and I believe that a very light comparation was made. In the introduction the authors indicated that there are many factors involved in concentration pollutants in the station: outdoor concentration and service time of subway (Song et al.); season, design of stations or tunnels, train frequency, density of passengers and ventilation systems (Martins et al); depth stations and season (Park et al.); while that the authors supposed the fully enclosed door also have effect. Therefore, more information about the station under study is necessary, and the comparison must be realized considered that differences. If there is not information about others stations those must be declared.
Lines 195-198. Due to the previously indicated, I believe that this declaration has not scientific support. How can the authors have declared that fully enclosed door really has an effect over behavior of pollutants if there are many factors that were not considered?
Lines 211-214. The authors declared that “the indoor PM concentrations were higher than the outdoor concentrations, which might have been affected by the concentrations of the previous day”, So, I would like to know why those days the inside PM could be influence for previous days and the other days no… Can the authors suggest some explanations? They give an explanation in their answer to previous review, but this is not possible to read in the manuscript.
Lines 219-220. The authors suggested the same explanations for SO2 indoor concentration on days 3 and 4 that for PM… So, I asked to me if those pollutants have the similar behavior. Moreover, do the authors believe that other factor could be affect the pollutant behavior in the subway station?
Lines 228-229. How can the authors explain that CO concentration is considerably lower indoor that outdoor, especially when is compared with the others pollutants? In the authors answer they suggest that this could be due to the HVAC systems, I believe that this idea must be develop in the manuscript, give more details about de HVAC system.
Finally, the main conclusion was that indoor pollutant concentrations were influenced for outdoor concentrations. Therefore, I don’t know the real contribution of this study, especially because there is a lack of information about the station characteristics and the discussion with other subway station is very weak, and it is complicated to evaluate the fully enclosed doors influence over the pollutants due to the lack of station description and possible effect of those characteristics over pollutant concentrations indoor.
Author Response
Response to Reviewer #1
Dear Reviewer#1,
Thank you again for the time and effort that you have put into reviewing the previous version of our manuscript entitled " Experimental investigation of air quality in a subway station with fully enclosed platform screen doors ". Your suggestions have no doubt of great help to our paper. All the changes have been highlighted in the revised manuscript.
Point1: I believe that authors did not really make substantially changes in the manuscript, especially in the discussion section. For instance, the comment: “Lines174-187. The authors compared their results with other references, but they only mentioned there is a different concentration in this study in comparation with other. However, I would like to know if the station conditions of those studies are different with this study. The references made their studies under the same ventilation systems or the station size or any information that allow to identify why that differences were registered”. The authors answer was: “The comparison has been enriched by further considering the effect of screen doors Unfortunately, the ventilation system and station size information given in previous studies was inadequate for a solid comparison”. But, in the manuscript only one sentences was added (lines: 195-198), and it have little scientific support. On the other hand, I can see in Table 5 that PM10 concentrations were very different Seoul [59] and Nanjing [58] but both have fully enclosed platform. So, how can the authors declare that fully enclosed platform could effectively prevent the airborne particulate matter?
Reply: Thank you for your valuable comments. We have tried to improve the interpretation of the daily trends shown in Fig. 3 and the comparison with previous studies in the discussion section, as highlighted in the revised manuscript.
The entrances of the tested Nanjing subway stations [58] were adjacent to traffic intersections. Therefore, the outdoor PM10 concentrations near the Nanjing subway stations could be highly increased due to the heavy traffic. Besides, their experiments were conducted in transitional season, when the ventilation systems instead of HVAC system were operated during the time of sampling. Therefore, the PM10 concentrations on the Nanjing subway platforms were higher compared with the concentrations on the Seoul subway platforms, though both having fully-enclosed screen doors [59].
As we discussed, the screen doors can prevent the PM generated by the train arriving at the platform, and more effective for fine particles than coarse particles. However, the indoor PM was mostly introduced from outdoors through the HVAC system and subway entrances, which could not be prevented by screen doors. We have clarified the statement about screen doors in the revised manuscript.
Point2: They only considered the simplest suggestions, but deeply changes did not include.
Reply: We have further improved the discussion section according to your comments, as highlighted in the revised manuscript.
Point3: Even, Reviewer 2 requested the authors that figures must be better explained. I believe that this is a requirement yet, especially with Figure 3. When I read the manuscript, I could not identify “the improvement in the interpretation of the data shown in these figures” according to the authors answered to Reviewer 2.
Reply: Thank you for your valuable comment. We have added more discussion about the variations of pollutant concentrations in the section 4.1 as highlighted in the revised manuscript.
Point4: Lines 43-44. Previously I requested to authors that they give us information about IAQ. What is IAQ mean? And Why is their inclusion important?
Reply: IAQ is the abbreviation of indoor air quality, which is a commonly-used term in the research field. The indoor environment of a subway station is typically confined and crowded, which can easily cause various types of air pollutants to accumulate in the indoor air, which will cause potential health risks to passengers. Therefore, it is important to study the IAQ of subway stations.
Point5: Lines 83-84. The authors give some station characteristics, but I believe that more information about the station is necessary. For instance: They must be indicated if the station is single or double tunnel, because as they mentioned this can have effect over pollutant concentrations (Moreno et al.), or station depth (Park et al.)
Reply: More information about the station has been added into the section 2.1.
The measured subway station was a non-transfer station with full enclosed platform screen doors. It is an underground station with two tracks in a single tunnel, which adopts a separated island platform design pattern with length and width of 120m and 14m, respectively (Figure 1 (a)).
Point6: Fig. 1a. This is a very useful information, but size and quality must be improved.
Reply: We have improved the image quality of Fig. 1a.
Point7: Line 126: The sentences: “There were two toilets in the subway, which resulted in a certain amount of NH3” must be translated to discussion because this an assumption, moreover it must be referenced.
Reply: According to your comment, we have transferred this description to the discussion section with a new reference.
- Zhang, K; Chen, X; Zhang, S; Wilson-Gray, B. Towards a Healthy Ride: Locating Public Toilets in the Shanghai Metro System. Applied Spatial Analysis and Policy. 2018, 11, 381-395.
Point8: Line 129: What is the HVAC mean? I can assume that correspond to air conditioning system.
Reply: HVAC is the abbreviation of the Heating, Ventilation and Air-conditioning.
Point9: Lines 137-138. Discussion
Reply: We have further discussed this sentence in the section 4.1.
Point10: Line 158: Discussion (Park et al.)
Reply: We have further discussed this sentence in the section 4.1.
Point11: Figure 3b. In the previous review, I requested to the authors explanation about the erratic behavior of some pollutants, for instance, CH2O showed a peak between 07:00 and 11:00 on day 3, and very different behavior on day 2 where the peak was registered between 18:00 to 22:00. The authors only answer in function of correlation between some pollutants (NO2, CO, PM2.5, PM10 and SO2) between outdoor and indoor concentration, but they don’t give explications about CH2O. This information must be added to discussion.
Reply: Indoor CH2O mainly came from the emissions of building materials, furniture and various adhesive coatings. We have added explications about the variations of CH2O in the section 4.1.
The indoor CH2O mainly accumulated from the emissions of building materials, furniture and various adhesive coatings [63]. The change of indoor CH2O concentrations could be related to indoor temperature. Higher indoor temperature can be helpful for the release of more CH2O from the building finishing materials [64]. This might explain the increase of CH2O concentrations occurred at the morning or evening peaks.
- Salthammer, Tunga. Formaldehyde sources, formaldehyde concentrations and air exchange rates in European housings. Building & Environment. 2019, 219-232.
- Liying Liu; Xiaoping Yu; Xiukun Dong; Qiang Wang; Yichao Wang; Jingjng Huang. The Research on Formaldehyde Concentration Distribution in New Decorated Residential Buildings.10TH INTERNATIONAL SYMPOSIUM ON HEATING, VENTILATION AND AIR CONDITIONING, ISHVAC. 2017, 1535-1541.
Point12: Figure 3. The authors indicated or justified the NO2, CO, PM2.5, PM10 and SO2 indoor concentration according to daily outdoor concentration, but they did not explain why some pollutants have a difference behavior between hours. For instance, why CO showed a peak early morning (08:00-11:00) on day 3, while that on day 2 the peak was registered at night (18:00-22:00), or How would the authors explain it that NO2 reached almost 0 ppb between 14:00 to 15:00 hours on day 3? Correlation analysis made by the authors do not answer this, and they neither answer them in the previous review. This information must be added to discussion.
Reply: It is a good suggestion. We have compared the variations of indoor air pollutant concentrations with their corresponding outdoor variations in the section 4.1, as highlighted in the revised manuscript.
Point13: Figure 3. The authors also did not answer about the peak of SO2 at weekend days. They said that SO2 is produced by is the combustion product of coal or oil and is mainly associated with industrial sources, in their review answers. I would expect that authors develop will develop deeply the hourly variation for the pollutants because they have a very fluctuating behavior. This information must be added to discussion.
Reply: The variations of indoor SO2 concentrations were basically correlated to the corresponding outdoor variations. The peak of indoor SO2 on the weekend was consistent to the increased outdoor SO2 during the same time period.
Point14: Lines 183-194. I keep my doubts with this discussion and I believe that a very light comparation was made. In the introduction the authors indicated that there are many factors involved in concentration pollutants in the station: outdoor concentration and service time of subway (Song et al.); season, design of stations or tunnels, train frequency, density of passengers and ventilation systems (Martins et al); depth stations and season (Park et al.); while that the authors supposed the fully enclosed door also have effect. Therefore, more information about the station under study is necessary, and the comparison must be realized considered that differences. If there is not information about others stations those must be declared.
Reply: We have added more information to strengthen the comparison, as highlighted in the section 4.2.
Point15: Lines 195-198. Due to the previously indicated, I believe that this declaration has not scientific support. How can the authors have declared that fully enclosed door really has an effect over behavior of pollutants if there are many factors that were not considered?
Reply: We have rephrased the statement about screen doors in the revised manuscript. The protection effect by the fully enclosed screen doors could be supported by the correlation analyses and the conclusions from previous studies. But it is worth noting that the indoor pollutants were still mostly introduced from outdoors through the HVAC system, which could not be prevented by screen doors.
Point16: Lines 211-214. The authors declared that “the indoor PM concentrations were higher than the outdoor concentrations, which might have been affected by the concentrations of the previous day”, So, I would like to know why those days the inside PM could be influence for previous days and the other days no… Can the authors suggest some explanations? They give an explanation in their answer to previous review, but this is not possible to read in the manuscript.
Reply: The average concentrations of indoor air pollutants (PM and SO2) on the subway platform were higher than the corresponding outdoor concentrations on days 3 and 4. The air conditioning system was switched off after the last train every day, possibly resulting in the accumulation of indoor air pollutants on the platform. Therefore, the average indoor air pollutants concentrations could be affected by the high concentration in the previous day, such as the day 2 in this study. We inferred that the concentrations of other pollutants on day 3 were also influenced by the high concentrations in the night of day 2.
Point17: Lines 219-220. The authors suggested the same explanations for SO2 indoor concentration on days 3 and 4 that for PM… So, I asked to me if those pollutants have the similar behavior. Moreover, do the authors believe that other factor could be affect the pollutant behavior in the subway station?
Reply: Please refer to the above reply to the point 16.
Point18: Lines 228-229. How can the authors explain that CO concentration is considerably lower indoor that outdoor, especially when is compared with the others pollutants? In the authors answer they suggest that this could be due to the HVAC systems, I believe that this idea must be develop in the manuscript, give more details about de HVAC system.
Reply: More details about the HVAC system has been given in the revised manuscript in the section 2.1. It is commonly seen that the indoor CO concentration is at very low level, because there is no indoor source. The indoor CO basically came from the traffic-contaminated air being brought down from street level, and has been diluted by the ventilation of HVAC system.
Point19: Finally, the main conclusion was that indoor pollutant concentrations were influenced for outdoor concentrations. Therefore, I don’t know the real contribution of this study, especially because there is a lack of information about the station characteristics and the discussion with other subway station is very weak, and it is complicated to evaluate the fully enclosed doors influence over the pollutants due to the lack of station description and possible effect of those characteristics over pollutant concentrations indoor.
Reply: We have to admit that this study has several limitations as you mentioned. We have improved the discussion with other stations in the revised manuscript. In sum, there are two contributions of this paper:
(1) In most previous studies, typically only one or two pollutants were measured in subway stations. In this current study, eight airborne pollutants were continuously measured for six days, which could provide more comprehensive indoor air quality data of a subway station. Further studies could be performed in more subway stations based on the research approach.
(2) In this study, the indoor-outdoor relationships have been discussed for five main atmospheric pollutants, while most previous studies focused on indoor/outdoor PM data. In addition, few studies have compared the daily variations of indoor and outdoor pollutants. This research could provide some reference for the I/O correlations of various pollutants in a typical subway station with a running HVAC system. The data could be useful for the comparisons by other studies.
Again, we are highly appreciative of your time and detailed review of our study. We believe we have addressed all of your comments appropriately. The manuscript is much improved.
Sincerely,
The authors
Reviewer 2 Report
The article has been improved and could be accepted if the authors address the following minor comment: the conclusions should improve the introductory part (the current format is very simple) and should include the future lines of research and the limitations of the study.
Author Response
Response to Reviewer #2
Dear Reviewer #2,
Thank you again for the great effort that you have put into reviewing the previous version of our manuscript entitled ‘Experimental investigation of air quality in a subway station with fully enclosed platform screen doors’. Your thoughtful points have enabled us to improve our work.
Point1: The conclusions should improve the introductory part (the current format is very simple) and should include the future lines of research and the limitations of the study.
Reply: Thank you for your valuable suggestion. We have revised the conclusion section as highlighted in the revised manuscript.
Again, we are highly appreciative of your time and detailed review of our study. We believe we have addressed all of your comments appropriately. The manuscript is much improved.
Sincerely,
The authors
Reviewer 3 Report
The manuscript has been improved and deserves publication.
Author Response
Dear Reviewer#3,
Thank you again for the time and effort that you have put into reviewing the previous version of our manuscript entitled " Experimental investigation of air quality in a subway station with fully enclosed platform screen doors ". Your suggestions have no doubt of great help to our paper. All the changes have been highlighted in the revised manuscript.